# An Introduction of a Modular Framework for Securing 5G Networks and Beyond

**Ed Kamya Kiyemba Edris** [1,*], **Mahdi Aiash** [1] **and Jonathan Loo** [2]

1   School of Science and Technology, Department of Computer Science, Middlesex University, London NW4 4BT, UK
2   School of Computing and Engineering, University of West London, London W5 5RF, UK
*   Correspondence: ee351@live.mdx.ac.uk

**Abstract:** Fifth Generation Mobile Network (5G) is a heterogeneous network in nature, made up of multiple systems and supported by different technologies. It will be supported by network services such as device-to-device (D2D) communications. This will enable the new use cases to provide access to other services within the network and from third-party service providers (SPs). End-users with their user equipment (UE) will be able to access services ubiquitously from multiple SPs that might share infrastructure and security management, whereby implementing security from one domain to another will be a challenge. This highlights a need for a new and effective security approach to address the security of such a complex system. This article proposes a network service security (NSS) modular framework for 5G and beyond that consists of different security levels of the network. It reviews the security issues of D2D communications in 5G, and it is used to address security issues that affect the users and SPs in an integrated and heterogeneous network such as the 5G enabled D2D communications network. The conceptual framework consists of a physical layer, network access, service and D2D security levels. Finally, it recommends security mechanisms to address the security issues at each level of the 5G-enabled D2D communications network.

**Keywords:** network services; access; security; framework; device to device; communication; 5G

## 1. Introduction

New use cases will be created and vertical industries supported in the Fifth Generation Mobile Network (5G). End-users will be able to access services at the edge from different service providers (SPs). The mobile network operator (MNO) and SP will also share infrastructure and security management while providing the service to the users. The Fifth Generation Mobile Network (5G) will also use network services such as device-to-device (D2D) communications as a underlay technology to push content to the edge [1,2]. The general security approach in solving security issues is using cryptographic techniques to achieve most security objectives. These cryptographic techniques should increase the reliability of security and privacy mechanisms in D2D communication, in form of anonymity, unlinkability, privacy, confidentiality, integrity, and authentication. These mechanisms should be lightweight due to mobile device computation and energy consumption constraints. In the past, D2D security was applied at the application layer, however, recently, network layer and physical layer security are new ways to achieve security objectives. For example, communication secrecy can be achieved at the lower layers without depending on higher layer encryption. With cryptographic methods such as the symmetric and asymmetric methods deployed at these layers, D2D communication security requirements can be achieved. Additionally, physical layer security can also be achieved by analysing and implementing the physical properties of wireless channels connecting D2D devices to provide secrecy capacity, channel-based key agreement, physical-layer authentication, and privacy-preserving anonymity [3].

There are a few security features in an information-centric networking (ICN) architecture. For example, to address the trust issue in ICN, a consumer can gain trust from the received content by deriving it from the publisher's credentials using their public key certificate [4]. In content-centric networking (CCN), data integrity is achieved by a publisher signing a piece of content with their private key and verifying the content using the publishers' public key. Data integrity is intrinsically provided in ICN by a named data object including a hash of data formatting. A symmetric encryption technique is used, and keys are securely distributed to authorised users, ensuring data secrecy. Due to name-based routing and the fact that the network's interests are dispersed throughout many areas, attacks such as denial of service (DoS) have less of an impact on CCN. Secure naming addresses content poisoning and cache pollution, which are basic security challenges that need to be addressed. Strong cache validation and self-certifying naming methods also prevent fraudulent content and unpopular content from spreading throughout the network [2]. Due to its capacity to support dynamic content objects and offer an effective content retrieval, self-certification has emerged as an effective strategy, particularly at the network edge of 5G technology [5].

The initial research to address the security issues of D2D communications led to the development of several security protocols to address security at different layers of communications. It was discovered that there is a need for a security framework that can capture all levels of the network. This article aimed to explain how these protocols could co-exist and interact within the context of a framework. Therefore, it explores the security of 5G and D2D communications and proposes a network service security (NSS) framework that includes a multi-level security model for a 5G-enabled D2D communication system. The NSS framework consists of three security levels that can be used to develop the underlying security protocols for an integrated security solution to provide privacy and security protection to network services at different levels of 5G-enabled D2D communications network.

The article's main contributions are summarised as follows:

- The security of 5G and D2D communications is explored by investigating the UE authentication and authorisation procedures;
- A security framework is proposed that addresses security at different levels of the network for D2D communications in 5G and beyond;
- The security model that applies different security mechanisms for network service delivery in 5G is explained;
- An integrated security solution for securing network services for 5G-enabled D2D communications by incorporating verified and evaluated security protocols in the proposed security framework is explored.

The rest of this article is structured as follows: an overview of the D2D security in the 5G network is presented in Section 2. While Section 3 introduces the designing and modelling of the security framework, an integrated security solution for D2D in 5G is presented in Section 4. This article is concluded in Section 5, with some recommendations.

## 2. Related Work

The Fifth Generation Mobile Network (5G) enabled D2D communication security challenges to be addressed using infrastructural and information-centric security mechanisms to protect devices, the communication channel and the network services. The New Generation NodeB (gNB) assists in establishing the connectivity of D2D devices, and is involved in the distribution of security information such as keys and certificates, which extends the decentralised security-centric methods into D2D architecture; however, the gNB acts as the trust authority.

*2.1. Security Mechanisms*

To achieve the main security objectives, which are authentication, authorisation and secure data sharing in a 5G-enabled D2D communication network, multiple security techniques are deployed.

Mobile networks adopted a service authorisation model that provides default services to every subscribed user, whereby implicit access authorisation is given to registered user equipment (UE) upon successful primary authentication. Service authorisation in legacy mobile networks, such as Fourth Generation (4G), was based on the static subscription of a user. Moreover, each UE's authorisation matrix is kept in the home network (HN) and then downloaded to the snetwork (SN) [6]. The SN then utilises the received permission matrix to grant the authenticated UE access to the services provided by the SP.

The standardisation and adoption of a static SP-based authorisation model have proven beneficial from an interoperability standpoint when applied to a market with a limited set of services supplied via wireless networks managed by one or two MNOs. In 5G, the UE will be authenticated to access the HN and authorised to access services in the HN and data network (DN) to support multiple shareholders. For new services, the authentication mechanism was decoupled from authorisation, and new authorisation processes were established. Network slicing provided by Network Function Virtualisation (NFV) and Software-Defined Networking (SDN) technologies is used in 5G to provide a diverse collection of services. A service authorisation architecture that allows the delivery of services from several infrastructure providers using an SP-based authorisation mechanism to enable existing implicit service authorisation while also protecting SPs from unauthorised service access is desirable because of the projected large range of service products and connected devices [7].

The authentication and authorisation mechanisms that were used in this research adopted Authentication Key Agreement (AKA) and access control methods to address security in 5G-enabled D2D communications networks. These protocols' security properties are derived from the security requirements of the system model which are secrecy, authentication, integrity, confidentiality and privacy [2].

2.1.1. Authentication

There are two authentication procedures specified in the 5G standard [6], i.e., primary authentication with two methods, namely 5G-AKA and Extensible Authentication Protocol (EAP)-AKA' and secondary authentication based on the EAP framework, which is an important step for 5G to become an open network platform. The UE and network authentication methods in 5G are classified as primary authentication. It is comparable to that used in the legacy systems, however, in 5G, the HN has been given more control during the authentication procedure. This procedure has an in-built home control, which allows the HN to be notified when the UE is authenticated in an SN and to make the final decision on mutual authentication with the UE, whether it agrees with the message exchange and verification process [6]. This applies to the authorisation process for non-3GPP technologies such as IEEE 802.11, due to it being independent of radio access technology. Secondary authentication provides secure communication between UE and DN outside the mobile operator domain. EAP-based authentication techniques and related credentials can be utilised for this. The UE can be authenticated with DN and obtain authorisation on establishing a data path from the operator network to DN, assisted by the HN Session Management Function (SMF). In this case, the DN could be a third-party SP. The DN might be providing data services such as operator services, Internet access or content services. The DN function has been mapped onto the third party domain in 5G architecture because of secondary authentication provided by DN Authentication Authorisation Accounting (AAA) servers [8]. In another applicable scenario, the HN might provide infrastructure services via network slices to other MNOs or SPs, even though they are in the same network domain; however, the service and security provision are handled by another party, therefore

secondary authentication could be applied to internal DN [9]. The primary and secondary authentications are discussed in detail in [10–13], respectively.

Mutual authentication is achieved when both parties confirm each other's identities and agree on a session key. The access security for the New Generation Radio Access Network (ngRAN) and 5G Core Network (5GC) involves mutual authentication between the HN and UE, key derivation for authentication, access network, non-access stratum, radio resource control security and non-3GPP access [6,10]. It provides integrity, ciphering and replay protection of signalling within the 5G network. The UE and 5G network mutual authentication rely on primary and secondary authentication procedures for accessing services in 5GC and from third-party SP/external DN, respectively. The 5G system supports mutual AKA between the UE and SN authorised by the HN, enabling the UE to securely access the HN via SN. The 5G-AKA or EAP-AKA' methods are mandatory for the 5G primary authentication procedure and the only authentication methods supported by UE and SN, for private networks' EAP framework, should be used as specified in [6] and as shown in Figure 1. The 5G-AKA and EAP-AKA' are discussed in [6,10,11,14,15].

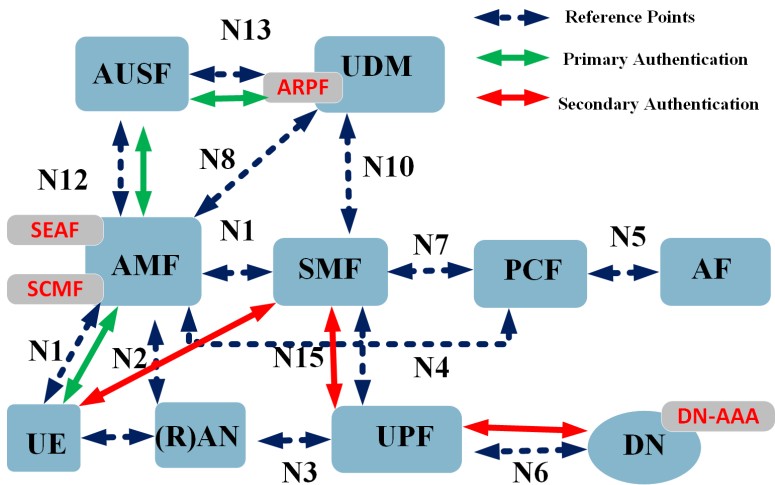

**Figure 1.** 5G security entities.

### 2.1.2. Authorisation

Mobile networks implicitly authorise service access after authentication. Generally, for authorisation, access control can be used to implement permission and access rights by protecting access to an object. When a subject wants to access, the subject's name is checked against a list; if it is on the list, then access is granted [16]. Conventional access control approaches to provide service access authorisation to the system have been proposed in related work and include Role-Based Access Control RBAC [17], Discretion Access Control (DAC) [18] and Attribute-Based Access Control (ABAC) [19]. Such access control mechanisms sometimes require additional techniques such as Encryption-based Access Control (EBAC) to provide a robust and efficient authorisation to complex systems including heterogeneous networks (HetNets). However, due to the complex characteristics of 5G, they are unable to provide a controllable and efficient mechanism to meet the criteria of 5G network service authorisation [12].

RBAC is a framework for specifying user access authorisation to resources, roles and responsibilities, and it follows principles such as the separation of duties, the least privilege and administrative activity segmentation. In contrast to ABAC, access control policies are developed by directly linking attributes with subjects. To achieve fine-grained access control, an efficient ABAC authorisation technique is employed based on user attributes and the access control authority grants the access rights.

Approaches based on Capability-Based Access Control (CBAC) have been suggested as a possible solution for the 5G network. CBAC uses an unforgeable token that designates

access to a resource to inform of abilities according to a set of rights [20]. Capabilities are a two-pronged method to access control, in which each subject is assigned to a capability list that specifies each object and the actions that the subject is authorised to perform on it. The access matrix is stored by row in the metadata of the object [16]. The subject presents a capability to the service server (SS) to obtain access to an object and the capability is transferable and non-forgeable. Local SPs could perform the CBAC, capability token validation and access right authorisation processes. This can be accomplished by locally implementing permission processes on distributed edge devices, making it feasible for D2D communications. Many access control systems for mobile network applications have adopted capability-based methods, but this has raised a few issues such as capability propagation and revocation [21].

With in-network caching, content objects may not always arrive from their original producer such as the SP, and content security cannot be considered in the traditional mobile network model based on secure and wireless or point-to-point channels [22]. This implies that content must be encrypted to prevent unauthorised access, invalid disclosure or modification by unauthorised parties using EBAC. By offering a framework for delivering access permissions to services, the existing access control mechanism reflects a good conceptualisation of authorisations. All these access control policies can be implemented independently or as an integrated access control solution.

The authorisation mechanism described in [6] uses the OAuth 2.0 framework as defined in RFC 6749 [23]. It states that client credentials should be used as grants and access tokens shall be in JavaScript Object Notation (JSON) web token format, which can be protected with JSON web signature in the form of a digital signature or message authentication code (MAC) built on JSON web signature [24].

2.1.3. Access Rights Delegation

Users can be assigned access permissions in the form of delegation, which is the process of assigning access rights to a user by either an administrative user or another user. The administrator user does not need to be able to use the access right, but a user delegation must be able to use the access right [25]. For authorisation and capability revocation management, a federated delegation method can be used in the capability development and propagation workflow. This could overcome issues in the access control strategy processing of a hybrid security mechanism by combining ABAC and CBAC with federated identity (FId) in a content-aware mobile network such as 5G [12]. Moreover, delegating some authentication and authorisation activities to other security domains facilitates 5G security policies and ubiquitous services access in different domains from multiple SPs. Processing capability validation in the HN and third-party SPs offers a D2D communications access control mechanism that is flexible, elastic, context-aware and fine-grained [26]. This inter-domain delegation and access authorisation enable 5G-enabled D2D communications security to be implemented beyond static authorisation.

In addition, the authors in [27] introduced a framework that proposed a self-delegation protocol for device authentication and proactive handover authentication using a delegated credential for unified network- and service-level authentication for wireless access. Two authentication and key agreement protocols were introduced as part of a security framework to secure transactions at the network and service levels in [28]. In a heterogeneous system such as 5G for multi-server collaboration, privacy protection is crucial, as presented in [29], so the authors used blockchain to develop heterogeneous multi-access edge computing (MEC) systems to offer privacy topology protection. The authors in [30] developed a privacy-preserved, incentive-compatible and spectrum-efficient framework based on blockchain that considers human-to-human spectrum utilisation and machine-to-machine communication. A framework for the Internet of Vehicles (IoV) architecture model and an authentication-based protocol for smart vehicular communication using 5G are both suggested in [31].

The comparison between some related work is shown in Table 1 in order to highlight the key differences between the other pre-existing security frameworks for heterogeneous networks security and the proposed conceptual framework in this research. It outlines the authors, their descriptions, and the variations among a number of criteria, including key hierarchy, protocols interface, privacy preservation, authentication, authorisation, single sign-on (SSO), formal verification and evaluation.

As discussed in the related work, some security features such as authentication, authorisation and permission delegation have been used in different D2D communications or 5G independently. However, there has been a lack of a framework that considered a multilayered security solution for a 5G mobile network including the D2D communication as a layer of the network. With 5G's unique characteristics, the promise of integration with the networks and pushing services to the edge, the proposed framework intends to provide an integrated security solution for D2D communications in 5G and beyond that is interoperable, verified and evaluated.

As discussed in the related works, some security features such as authentication, authorisation and permission delegation have been used in different D2D communications or 5G independently. However, there has been a lack of a framework that considered a multilayered security solution for 5G mobile networks, including D2D communications as a layer of the network architecture. With 5G's unique characteristics, the promise of integration with the networks and pushing services to the edge, the proposed framework intends to provide an integrated security solution for 5G and Beyond that is interoperable, verified and evaluated.

**Table 1.** Comparison of Related Work Based on the Proposed Conceptual Framework.

| Authors | Year | Description | Differences | | | | | | |
|---------|------|-------------|:-:|:-:|:-:|:-:|:-:|:-:|:-:|
| | | | 1 | 2 | 3 | 4 | 5 | 6 | 7 |
| [27] | 2004 | proposed a framework with self-delegation and handover protocols | ✓ | ✓ | ✓ | x | x | x | x |
| [28] | 2014 | introduced a framework with two AKA security protocols | ✓ | ✓ | ✓ | x | ✓ | ✓ | x |
| [29] | 2020 | presented a blockchain for privacy preservation in MEC systems | ✓ | ✓ | x | x | x | x | ✓ |
| [30] | 2020 | presented a framework that provides privacy preservation and is spectrum-efficient | ✓ | ✓ | x | x | x | x | x |
| [31] | 2020 | proposed a framework for the IoV architecture model and an authentication protocol | ✓ | ✓ | x | x | x | x | ✓ |
| Proposed in this paper | . | proposes conceptual framework with 5 security protocols for 5G communications | ✓ | ✓ | ✓ | ✓ | ✓ | ✓ | ✓ |

Parameters—1: privacy preservation; 2: authentication; 3: authorisation; 4: SSO; 5: key hierarchy and interface; 6: formal verification; 7: performance evaluation
Notations—✓: considered; and x: not considered

## 3. Proposed Network Service Security (NSS) Framework

This section presents the proposed security solution and explains how protocols coexist and interact with each other in the context of the framework. This article adopts a network service abstraction concept from [32], the system architecture from [33] and the security architecture from [6]. The UE registers with HN and starts receiving roaming services from a visiting network (VN). The network services consist of services that rely

on other services, such as D2D communications and ICN. It would require integrating 5G with other network architectures such as ICN, content delivery network (CDN) and cloud computing [2,34]. The UE would register with MNO or SP and subscribe for such services, allowing it to connect to the network and access services from HN and other SPs in different domains.

After being authenticated to access the network, the UE may need to execute a secondary authentication with the SP, which authorises the UE to access its services as well as authorising the UE to perform other activities such as data caching and sharing with other UEs. D2D communications can be deployed to support different use cases such as traffic offloading, location-based services and vehicle to everything (V2X) communications. In addition, D2D direct communication could be established for content sharing and gaming, therefore the communication must be established using a secure and efficient method with the minimal involvement of the gNB [2,26].

The MNO/SP controls the service subscription, access and content retrieval authorisation as well as enables the normal service operation of the cellular network. However, the SP in 5G could be the MNO, third party, or another SP that uses the MNO's infrastructure as a tenant via network slicing [26]. The gNB controls the UE in cellular coverage and communications between two devices, whilst the D2D devices control the UE out of coverage scenario. Moreover, the MNO is in charge of the user's network access, connection setup, resource allocation and security management. The MNO/SP may block the UE from accessing the services or hide their visibility. To deliver inter-operator D2D services, various networks should sign an inter-operator agreement. The communication channels between UE and networks as well as the D2D devices are all susceptible to attacks, as HN and VN may also be interested in eavesdropping on D2D communications [35]. It should be mentioned that the content access and retrieval process, which include content discovery and distribution, were also considered in this study.

In this article, an approach that integrates both infrastructural-centric and information-centric security services is proposed. The hybrid security framework will focus on:

- Information-centric security services—providing data confidentiality, integrity and availability;
- Infrastructural-centric security services—providing entities authentication, access controls of the user to network and services.

### 3.1. Network Service Security Architecture

To address the security threats in 5G-enabled D2D communication and to provide the secure delivery of network services, the proposed security framework assumes that network access security has been achieved. The main concern is the secure access of services by the UE from the SP. The UE should be able to obtain authorisation to access and share the data with another UE, hence achieving service security. The verification of the authenticity, integrity and provenance of the named data object against the producer must be performed before the UE is granted access. Another concern is whether the right data are being published and can be restricted even during out of coverage. The UE also should be able to share data without involving the HN even during coverage, which addresses one of the D2D security problems.

After a successful authentication procedure with the network, the UE can request access to services of SP via HN, and the SP verifies the UE and grants access to UE. Security is implemented by various security mechanisms, which should be interoperable with each other. Before addressing security at the different levels, these levels must be defined and the security model function of each level has to be specified.

To address the security issues of accessing the network and services, a unified modular architecture is required, as shown in Figure 2, to support the proposed security framework, and the generic architecture consists of the following security entities which are modified according to various security models:

- UE: The end-user's device that is trying to access the services;

- HN AAA servers consist of security anchor function (SEAF) as SN or authenticator, Authentication Server Function (AUSF) for authentication from the HN if the service is in a home-controlled environment. From a network perspective, the Authentication Credential Repository and Processing Function (ARPF) maintains keys and other security contexts that are utilised for primary authentication [6];
- External AAA servers for service authorisation and secondary authentication in local and external DNs authenticate and authorise the UE to access the service and permission delegation to share with other UE;
- SS: A server storing the content/services that the UE is trying to access which could be controlled by either the HN as an internal SP or controlled by an external SP.

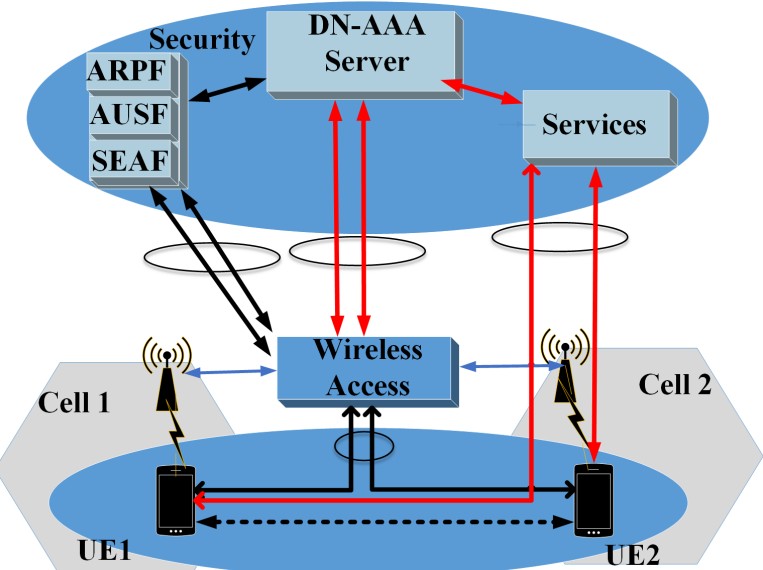

**Figure 2.** Unified modular architecture.

### 3.2. Security Modelling

This subsection discusses security modelling and a multilayered approach in a modular framework. The secure services model allows the UE to securely access network services from the network and SP. The wireless network is under the control of the 5G network, whose root certificate is kept in the Universal Subscriber Identity Module (USIM) [6]. In the 3GPP standard, it is mentioned that the UE and ARPF share long-term K by performing a challenge and response, the shared key with other information including identification information is used by the ARPF to generate session keys through collaboration with SEAF and AUSF as explained in [10]. The AUSF is in charge of the network and UE mutual authentication; after initial authentication, the communication link with HN gNB is left open until it is disconnected. If the UE moves to a different gNB, it will require re-authentication and the generated keys and security information might be reused during the handover—which is handled by the ARPF in both the HN and VN, this is out of the scope of this work.

In this case, the UE obtains access to the services after connecting to the wireless network, 5G is based on a security architecture that is host-centric and the CCN is information-centric; hence, the hybrid approach of the security framework. Normally, the security of the content relies on the encryption of the content object as the producer must register the content object to the database owned by the SP achieving the validation and authentication of the named data object. Multiple methods are used to address the complexity of this system model.

The security modelling in Figure 3 leverages the security principles by 3GPP [6] by applying the authentication and authorisation methods that grant the UE service access and permission to engage with other UEs. A multilevel framework is proposed to align

with the network service abstraction and 5G protocol stack [32]. It comprises Network Access Security (NAC), Service Level Security (SLS) and D2D Security (DDS) levels which are parallel with the network service abstraction. It also explores physical layer security (PLS) to highlight the need for security at each level and integrated solutions. This security framework's novelty is mainly on SLS and DDS, as PLS and the NAC have been extensively studied by related work. The PLS and NAC provide security for physical and network access; the SLS provides security for services access; and DDS provides security for D2D services sharing.

The framework considers the security link between all the security levels, and in addition to the security entities, it is represented as a unified security model. This security framework intends to provide the UE with secure service access and the sharing of these services with UE from another network without losing their initial network access. The security framework intends to provide secure communication and sharing between UEs without the need for HN as the central authority.

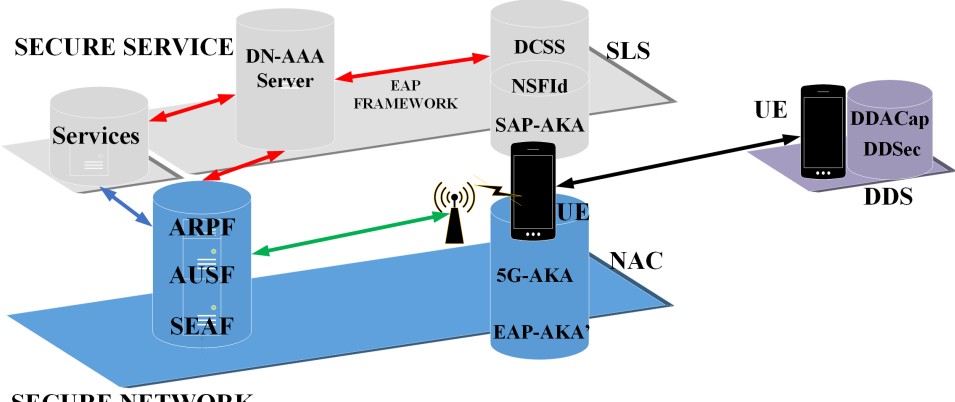

**Figure 3.** Security framework.

### 3.2.1. Physical Layer Security

This level of security is concerned with the PLS such as the spectrum, resource allocation, interference and signal. Even though PLS is out of scope, this article gives a brief overview of PLS solutions applied by relying on wireless channel characteristics such as interference, signal and fading—whilst the quality of the attacker's signal can be degraded through keyless secure communication by using signal engineering and processing techniques. Different studies explored PLS in 5G, and a detailed review of different PLS techniques are presented [36–38]. For example, artificial noise injection is used to increase channel quality and an anti-eavesdropping signal approaches are used to align multiple users' signals at the eavesdroppers. Secure beamforming, on the other hand, improves the spatial distribution properties of the transmitted signal, resulting in a greater difference in channel quality between legitimate users and eavesdroppers.

### 3.2.2. Network Access Security

The NAC was well-defined by 3GPP in 5G security standards and has been extensively studied in various related works [10,14,39–44]. The 3GPP specifies that, for UE to access the network, it requires a primary authentication process, which addresses the NAC. This includes the AKA protocol, which enables the UE and HN to authenticate each other.

Mobile subscribers will be able to access network services through ngRAN using their UEs, taking advantage of a variety of wireless communication technologies. As a result, secure access is critical to the 5G principle design, and 3GPP has defined the security requirements in [6] as well as the system architecture in [45] to support its objectives. The UE's connections should be secured by 3GPP's standardised security mechanisms. Both subscribers and MNOs require these mechanisms to provide security guarantees, such as

the authentication and trust of involved parties, as well as the confidentiality and integrity of the user's data. For the UE to access the network, the UE and the network must mutually authenticate, and then the UE must further authenticate using the security context to access services provided by the SP, which might be the same network provider or a third party via the DN function. Through layered access and security, the UE can gain access to network services. The AKA protocol mutually authenticates the UE and the HN and establishes a session key for UE and SN in which they can have secure communication over a wireless channel to provide network access security. As mentioned earlier, the 5G standard recommends 5G-AKA and EAP-AKA' protocols as preferred methods for primary authentication to address the most significant security requirements in 5G [6].

The network-level authentication is responsible for verifying that the UE has access to the right network when it connects to it [6]. In 5G, the messages between the UE and the access network via radio interface are to be encrypted. The security at the network level was extensively investigated in [39,42]. Furthermore, under the 3GPP definition, the AKA messages were implemented according to a security standard which outlines certain security properties that must be met. The security context obtained from the authentication on this level can be utilised for further authentication when the UE wants access to other services. The security on this level is concerned with authenticating devices in the network and mobility of the UE. In addition, the handover authentication requires the UE and new AN to re-authenticate both parties. To access the network services provided by the MNO, the UE and the network must mutually authenticate each other to establish a secure channel of communication, trust and the authenticity of the device in the network. After this level of security assurance, the UE can request to access other services from the servers in the CN or from third party service provider.

### 3.2.3. Service Level Security

The user must be verified to use the services at this level, and it must be verified whether the user is attempting to access the services with the right permission. As previously stated, this research is focusing on SLS due to new emerging services being promised in 5G, in addition to PLS and NAC having been extensively investigated. With 5G extending the mobile network's potential through the use of additional resources, dense connection, and the enabling of vertical industries, service provision is getting more challenging, especially from a security standpoint. However, the NAC is revisited in the discussion of SLS, as both the UE and network must be mutually authenticated for the UE to join the network. Security on the service level is concerned with authentication and authorisation between the UE, MNO and SP, which gives the UE access to the services and SP the ability to provide the services securely [26].

Since 5G is a large-scale HetNet in nature, some studies on service security are still relevant to this research. The service level protocol for the mobile network in [27,28] addressed security concerns when the UE is accessing services provided by the SP, this was based on IP-based networks and future networks. In [46], the authors proposed an open architecture based-service level protocol for mutual authentication between the UE and SP. After establishing connectivity to the network, the UE must be authenticated and authorised to use the services, which is addressed by SLS [12]. SLS also requires mutual authentication between the UE and the SP to establish a secure communication channel, with a focus on zero trust [11].

### 3.2.4. Device-to-Device Security

D2D communications' security was addressed to some extent in 4G and has to be explored more in 5G as it uses D2D communications as an underlay technology critical to its functionality and attaining its key objectives [1,3,32,47,48]. This study is concerned with service security and how the existing D2D security can be improved. How will the UE deal with the data accessed after being granted access to the network and service? To date, D2D authentication and authorisation have been dependent on various security

procedures, requiring it to be authenticated and authorised each time it disconnects from the network. Furthermore, in out-of-coverage conditions, the UE is currently not able to share restricted data with another UE. What happens to data on the UE and how can it be securely exchanged with or without network support is the subject addressed in this study. As a result, the DDS security level will attempt to address these concerns [2,26].

### 3.3. The Protocols

These security protocols that are defined in the framework to address security on different levels of the system model, as shown in Figure 4, were verified for security guarantees and evaluated for performance effectiveness. The framework incorporates various security protocols to provide an integrated security solution to a 5G-enabled D2D communication network. The protocols are formally analysed in related work, and they are as follows:

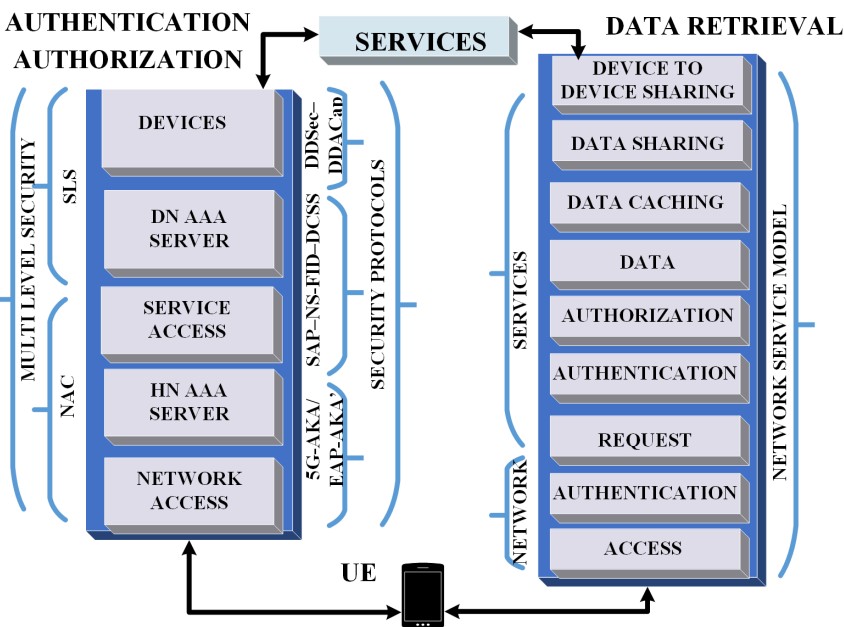

**Figure 4.** Security Model.

3.3.1. Network Access Security

NAC protocols and related work are explored in [10,15].

- 5G-AKA Protocol: enables the UE and the HN to establish mutual authentication and anchor keys [10].
- EAP-AKA' Protocol: enables the UE and the HN to establish mutual authentication and anchor keys [15].

3.3.2. Service Security

SLS protocols and related work are presented in [11–13].

- Secondary Authentication Protocol (SAP)-AKA Protocol: enables the UE and the SP to establish mutual authentication and anchor keys [11].
- Network Service-Federated Identity (NS-FId) Protocol: enables the UE and the SP to achieve mutual federated authentication and authorisation [12].
- Data Caching and Sharing security (DCSS) Protocol: allows the UE to cache and share data accessed from the SS [13].

3.3.3. Device-to-Device Security

DDS protocols and related work are presented in [26].

- Device-to-Device Service Security (DDSec) Protocol: provides authentication and authorisation to share the cached data between two UEs in proximity, with network assistance.
- Device-to-Device Attribute and Capability (DDACap) Protocol: provides security authentication and authorisation to share the cached data between two UEs in proximity without network assistance.

### 3.4. Formal Verifications and Performance Evaluation of the Security Protocols

These protocols are formally verified for security guarantees that align with the security requirements of the unified modular architecture in Figure 2, as demonstrated in [10–13,26].

### 3.4.1. Formal Verification Approach

Formal methods and automated verification were applied to security protocols such as AKA that provide weak assurances due to the use of strong abstractions, protocol simplifications and limitations in the properties' interpretation. To give solid guarantees, formal approaches were already used to examine security protocols in [12,14,15,43]. Most verification approaches and tools struggle with security protocol features such as those employed in the proposed framework. This is due to the use of cryptographic primitives such as the sequence number (SQN) and exclusive-OR (XOR), which have algebraic features that make symbolic reasoning difficult [14]. As a result, some tools are incompatible with manual proof checks. Many automated verification tools can be used for this security analysis, including Automated Validation of Internet Security Protocols and Applications (AVISPA) [49], Tamarin [50], and ProVerif [51].

### 3.4.2. Performance Evaluation Based on Analytical and Simulation Approaches

To check the effectiveness of these protocols, performance evaluation was carried out using analytical and simulation methods presented in [52]. The analytical model associates an enhanced label to each communication and each decryption based on the ProVerif and Applied pi-calculus processes used in the verification of the protocol.

The performance parameters and metrics are displayed in Tables 2 and 3 and the outcomes for each level are represented by applicable protocols in Table 4 for an analytical model.

The simulation model is built on the NS-3 5G mmWave module [53,54] to replicate the current non-standalone deployment of 5G using 5G radio technologies and the LTE network. By evaluating efficiency, throughput and computational cost, the communication and processing costs associated with the protocols are taken into account.

The performance parameters and metrics are displayed in Table 5 for computational cost, Tables 6 and 7 for communication cost and Tables 8 and 9 for the simulation model results at each level with a specific protocol.

**Table 2.** Cost Description.

| Term | Description |
|------|-------------|
| $n$ | size of the message |
| $m_i$ | size of the $i$th encryption |
| $e$ | cost of unitary encryption |
| $d$ | cost of unitary decryption |
| $s$ | cost of unitary output |
| $l_i$ | label in relation to the state |
| $c_i$ | cost in relation to the label |

**Table 3.** Metrics Variables.

| Variable | Description |
|----------|-------------|
| a | s + e |
| b | 2s + 2e |
| c | 3s + 3e |
| f | 4s + 4e |
| g | 5s + 5e |
| h | 6s + 6e |
| i | 7s + 7e |

**Table 4.** Protocols Performance Evaluation.

| Protocols | Efficiency | Throughput |
|-----------|-----------|------------|
| 5G Protocols | | |
| 3GPP-5G-AKA | $\frac{3GPPP-5G-AKA}{3d}$ | $\frac{d}{21s+5e+5d}$ |
| SAP-AKA | $\frac{A}{8d}$ | $\frac{3d}{20s+19e+19d}$ |
| NS-FId | $\frac{B}{10d}$ | $\frac{2d}{28s+28e+28d}$ |
| DCSS | $\frac{C}{8d}$ | $\frac{2d}{20s+20e+20d}$ |
| 5G D2D Protocols | | |
| DDSec | $\frac{D}{7d}$ | $\frac{4d}{27s+27e+27d}$ |
| DDAcap | $\frac{E}{4d}$ | $\frac{4d}{22s+19e+19d}$ |

**Table 5.** Approximate Time for Cryptographic Operations.

| Notation | Description (Time to Compute) | Rough Computation Time (ms) |
|----------|------------------------------|------------------------------|
| $T_{Av}$ | authentication vectors | 33.5 |
| $T_h$ | hash function | 5 |
| $T_{Se}$ | symmetric encryption | 4 |
| $T_{Sd}$ | symmetric decryption | 5.5 |
| $T_{Ae}$ | asymmetric encryption | 8 |
| $T_{Ad}$ | asymmetric decryption | 9.5 |
| $T_{Tn}$ | token | 5 |
| $T_{Ts}$ | timestamp | 5 |
| $T_{KDF}$ | NAC/SL key | 12.0 |
| $T_{D2D}$ | D2D key | 20.0 |
| $T_E$ | execute | 21.5 |
| $T_V$ | verify | 12.5 |

Furthermore, the security formal verification and performance evaluation approaches of this framework's underlining protocols were extensively explored in [10–13,26,52], respectively. The behaviour and cost of an algorithm are affected by the cryptographic scheme, formally analysed security properties and system model. An example of this is the employment of symmetric or asymmetric cryptography in a mobile network with a variety of stakeholders from various security realms. As a result, utilising this method

with this conceptual framework makes the important cost aspects obvious, helps in the development of security protocols and aids in the selection of cost-effective solutions. Other communications systems in addition to mobile networks can also use these techniques.

**Table 6.** Cryptographic Primitive Size.

| Primitive | Value |
|---|---|
| Symmetric key | 128 bits |
| Asymmetric key | 256 bits |
| SHA256 | 256 bits |
| Token | 128 bits |
| Nonce | 128 bits |
| 5G IDs | 64 bits |
| D2D IDs | 256 bits |
| Nonce key | 256 bits |
| D1 | 256 bits |
| Strings | 32 bits |
| MAC | 64 bits |
| SQN | 48 bits |
| Timestamp | 16 bits |
| RES | 256 bits |

**Table 7.** Evaluation Metrics

| Parameters | Values |
|---|---|
| Throughput | bits/ms |
| Latency | ms |
| $m$ | messages primitive cost |
| $n$ | total sum of $m$ |

**Table 8.** Computational Cost of the Protocols.

| Protocols | Computational Time (ms) | Total Time |
|---|---|---|
| | **(ms)** | **(ms)** |
| 5G Protocols | | |
| SAP-AKA | $T_E + 6T_{Se} + T_{Ae} + 6T_{Sd} + T_{Ad} + 1T_{Av} + 13T_{KDF} + 6T_V$ | 439.2 |
| NS-FId | $T_E + 6T_{Se} + 4T_{Ae} + 6T_{Sd} + 4T_{Ad} + 1T_{Av} + 2T_{KDF} + 5T_h + 2T_n + 10T_V$ | 396 |
| DCSS | $T_E + 8T_{Se} + 8T_{Sd} + 4T_h + 2T_n + 10T_V$ | 285 |
| 5G-AKA | $T_E + 6T_{Se} + T_{Ae} + 6T_{Sd} + T_{Ad} + 1T_{Av} + 8T_{KDF} + 2T_h + 11T_V$ | 545.5 |
| 5G D2D Protocols | | |
| DDSec | $T_E + 4T_{Se} + 3T_{Ae} + 4T_{Sd} + 3T_{Ad} + 2T_{K_{D2D}} + 7T_h + 1T_n + 2T_s + 7T_V$ | 360.5 |
| DDACap | $T_E + 5T_{Ae} + 5T_{Ad} + 2T_{K_{D2D}} + 7T_h + 1T_s + 8T_V$ | 307.5 |

**Table 9.** Communicational Cost of the Protocols based on Simulation Modelling.

| Protocols | Total Communication Cost (Bits) ($n$) | Number of Messages ($m$) |
|---|---|---|
| 5G Protocols | | |
| SAP-AKA | 3136 | 9 |
| NS-FId | 5472 | 10 |
| DCSS | 4096 | 8 |
| 5G-AKA | 5898 | 10 |
| 5G D2D Protocols | | |
| DDSec | 7904 | 7 |
| DDACap | 5760 | 5 |

## 4. An Integrated Security Solution

The inclusion of security features in the suggested security framework is covered in this section. The proposed NSS framework covers security for network services in 5G-enabled D2D communications from the point at which a UE requests access to the network via wireless access to the point at which it is allowed to share the service with another UE. The goal of the NSS framework is to defend against the dangers described in [2] by protecting the entities participating in communication and the data being communicated over communication channels in various security domains and scenarios. The NAC, SLS and DDS levels of the solution are as follows:

- NAC provides primary authentication and is concerned with the security of the 5G access network. It safeguards the entities, the wireless data connection and the correspondence between the UE, SN and HN.
- SLS offers secondary authentication and authorisation and is concerned with UE service authorisation. It safeguards information, entities and communication between the UE, HN and SP in several domains.
- DDS addresses the security of D2D communication, enabling data sharing in both network-assisted and non-network-assisted communication and offers authentication and authorisation between two UEs that are close to one another. Data, entities and communication between two UEs and across networks are all protected.

An integrated security framework can be created by including some solutions into this safe framework. Each level of the security model is made up of security protocols in the following linked work. In order to provide security with the NAC, 3GPP standardised 5G-AKA and EAP-AKA [6,10]. The authors [11–13] proposed protocols that deal with security at the service level of 5G networks. In [26], the authors proposed investigating D2D communication security and proposed two security protocols that offer security covering many D2D scenarios.

These security framework's underlying protocols address security on the network, service and D2D levels of communications. As illustrated in Figure 5, these security levels are connected by the protocol interfaces, and the protocols are encapsulated while addressing the security requirements from one level to another. These protocols employ or share some security contexts in order to provide an integrated security solution; nevertheless, this should not compromise security on another level or domain. Additionally, Figure 6 shows a hierarchy of keys, and explains how keys are generated and shared in various contexts.

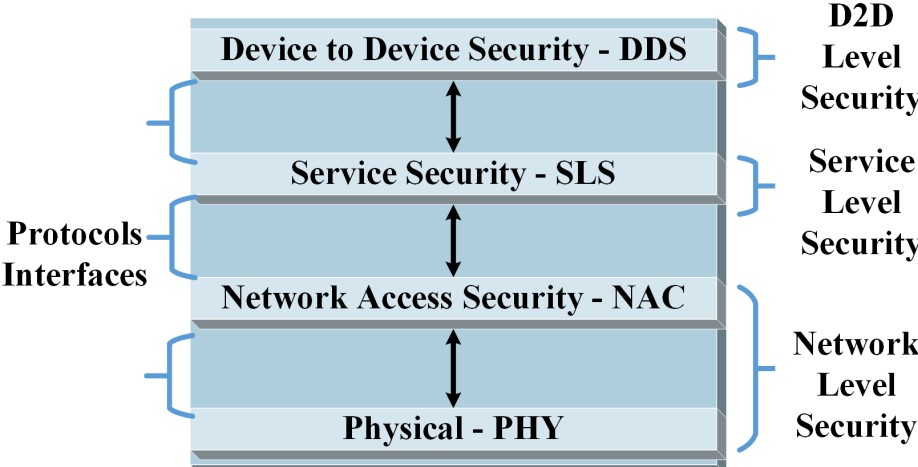

**Figure 5.** Interface of the Security Protocols.

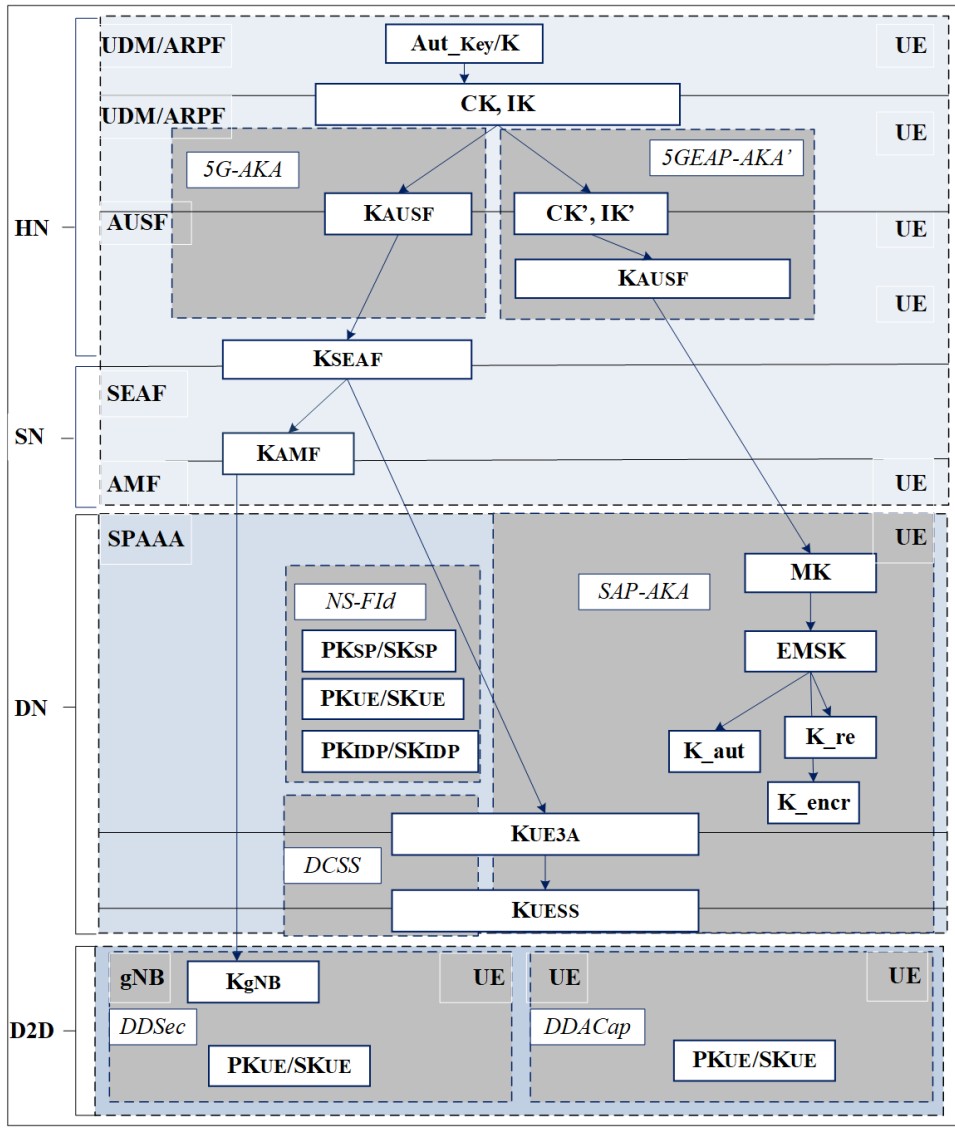

**Figure 6.** Key Hierarchy and Derivation.

*4.1. Connection of Different Levels of NSS Model in 5G Enabled D2D System*

When a user requests network access using a UE, the closest SEAF will start the primary authentication procedure. In [32], the procedure for the UE to access the network and services is covered. According to the following descriptions, the security model aims to provide three secure connections between the UE and HN; UE and SP; and UE and UE at various phases:

- In phase 1, a primary authentication protocol is initiated when the UE presents a network access request. According to [6,10], a 5G-AKA or EAP-AKA' protocol can be selected as the AKA procedure between UE and HN.
- In phase 2, after primary authentication, the UE submits a service request that initiates a secondary authentication or authorisation process, which is handled by SMF in the HN network and SP AAA in the SP network. Depending on the registration status and security guidelines outlined in [11,12], the SAP-AKA or NS-FId protocols may be utilised at this time.
- In phase 3, UE sends a request for data caching and sharing authorisation after getting access to the services using the DCSS protocol described in [13].
- In phase 4, once the UE was given permission to cache and share data, it can publish that data by broadcasting the data name to other nearby UEs, in this example, $UE_A$ and $UE_B$. The DDSec and DDACap protocols are invoked by another UE after an interested UE sends it the request, as defined in [26].

*4.2. Federated Security in 5G*

This section covers the integration of federated security into 5G as well as how the UE performs SSO. The benefits of adopting FId in mobile communications and how it eliminates the need for the UE to continually authenticate and authorise services, including in roaming scenarios, were presented in [55], which also discussed federated identity management in 5G. When the suggested solution in this article is implemented in a 5G-enabled D2D communications network, the tokens and caching data of the security processes are reduced when the UE needs to re-authenticate to the network or perform handover authentication while roaming but due to SSO, tokens and caching data of the security processes are reduced.

Next, the following steps will show how federated security is used in 5G communication:

- Step 1: Following network authentication, the UE requests the SP for service authorisation;
- Step 2: The UE is forwarded by the SP through SMF to the identity provider (IdP), which creates and assigns the FId to the UE;
- Step 3: Using the UE's identity token, the IdP and UE carry out federated authentication operations.
- Step 4: The UE uses an identity token to ask the SP for an access token, which the SP-AAA then issues along with a refresh token. SSO has now been accomplished;
- Step 5: If the access token is legitimate, the SS will grant the UE's request for access to the service;
- Step 6: Using the cached access token, the UE requests caching and sharing authorisation with other UE after getting access to the service.

An integrated security solution addressing security risks at various levels of the system model is provided by the interface between the underlying security protocols of the security framework using supported security context, as shown in Figure 7.

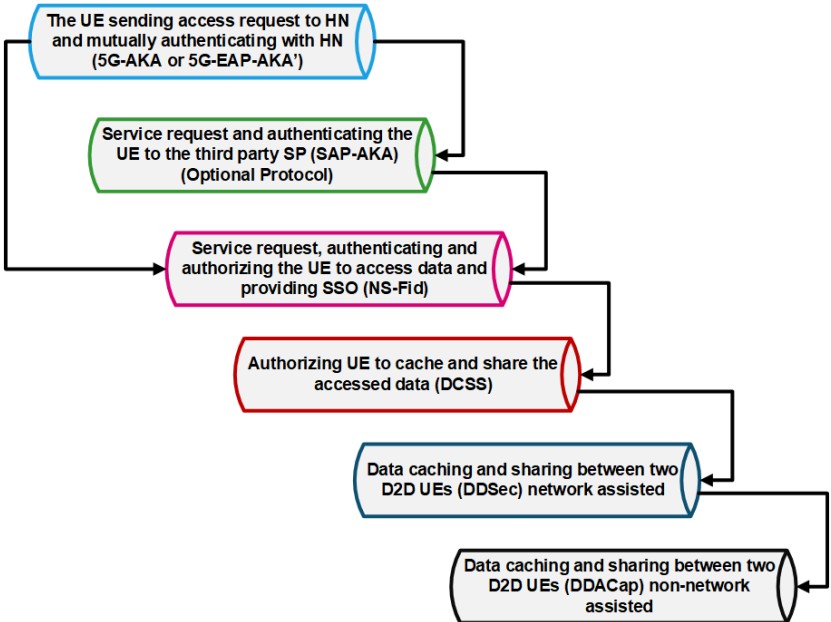

**Figure 7.** Integrated Security Solution.

## 5. Conclusions

A D2D communications network with 5G capability is made up of several systems backed by different technologies. This prompted the creation of several security solutions that deal with security for particular applications or layers. In addition, new use cases for accessing other services from SPs in various domains are made possible by 5G network services, posing additional security difficulties. Consequently, a comprehensive security solution that handles these problems is required. End-users will frequently access services from several SPs using their UE, which will present a new security risk. Additionally, infrastructure and security management may be shared amongst SPs. This article examined security in D2D communications, 5G, and beyond. It described the system's levels, including those that have been addressed in related works and those which still require attention. A security framework was proposed that outlined the security levels and entities involved in the authentication and authorisation processes. For the proposed security framework, it established the underlying security protocols, which were fully verified for security guarantees and appraised for efficiency. As part of an integrated security solution, the designed underlying security protocols using the suggested framework were streamlined to certain system levels and solved various security challenges. The protocols, however, can be used as a standalone solution, sharing some security context to comply with 5G security standards while permitting interoperability with third-party solutions without compromising security at any level.

Current studies have focused on developing security solutions that apply to one layer without considering the security of the layer below or above. The future direction of the research in this article is to extend the framework's application to the next-generation mobile network and other systems such as Internet of Things (IoT) and autonomous vehicles. This framework could develop security mechanisms and evaluate their effectiveness and interoperability as an integrated solution.

**Author Contributions:** Conceptualisation, E.K.K.E., M.A. and J.L.; methodology, E.K.K.E. and M.A.; validation, E.K.K.E. and M.A.; investigation, E.K.K.E.; writing—original draft preparation, E.K.K.E.; writing—review and editing, E.K.K.E. and M.A.; supervision, M.A. and J.L. All authors have read and agreed to the published version of the manuscript.

**Funding:** This research received no external funding.

**Conflicts of Interest:** The authors declare no conflict of interest.

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
