# Peer review of "An Introduction of a Modular Framework for Securing 5G Networks and Beyond"

_2673-8732, doi:10.3390/network2030026_

Round 1

Reviewer 1 Report

The paper is well drafted and it is original. The papers recommends security mechanisms to address the security issues at each level of the 5G enabled D2D communications network.

The authors are addressed various security mechanisms and issues. It is advised authors to tabulate the performance parameters at each level of security mechanism of 5G enabled D2D network in the current or existing or recent literature findings.

Author Response

Hi reviewer, thank you for your comments and feedback, we have added parameters and metrics in a tabular form used in evaluating underlying protocols of the proposed security framework and related work, they can be found between page 12 and page 14 of the manuscript. We have also corrected the grammar mistakes throughout the paper.

Reviewer 2 Report

This paper discusses 'An Introduction of a Modular Framework for Securing 5G Networks and Beyond'. By and large, the paper is very well written and the idea presented in this paper is new. This article presents a modular structure for network service security (NSS) for 5G and beyond, which comprises various network security degrees. In 5G enabled D2D communications, it examines security challenges affecting users and service providers (SPs) in an integrated and heterogeneous network. A physical layer, network access, service, and D2D security levels make up the conceptual framework. Finally, it recommends security techniques to solve the security concerns at each level of the 5G-enabled D2D communication network. The paper is well written, however, there are few minor issues:

1. Please be consistent, for example, message authentication code (MAC) and Device to Device (D2D). Either use a small or capital alphabet when defining an abbreviation.
2. Future directions are missing. 

Author Response

Hi reviewer, thank you for comments and feedback, we have corrected abbreviation definitions by using capital letters throughout the manuscript. We also added future direction of the research in the conclusion section. We have also corrected the grammar mistakes throughout the paper.

Reviewer 3 Report

To enhance the 5G secure, the authors propose a network service security modular framework for 5G and beyond that consists of different security levels of the network, which reviews the security issues of D2D communications in 5G. It shows security mechanisms to address the security issues at each level of the 5G enabled D2D communications network. This paper is meaningful for 5G network and secure. There are some comments below which I recommend to give one chance to take a revision. A more comprehensive literature survey may be provided and compared with distributed blockchain-based trusted multidomain collaboration for mobile edge computing in 5G and beyond, BLCS: brain-like distributed control security in cyber physical systems, blockchain-enabled tripartite anonymous identification trusted service provisioning in industrial IoT. Also, more results should add in the revision. 

Author Response

Hi reviewer, thank you for your comments and feedback, we have added the results from evaluation of the underlying security protocols of the security framework to effectiveness. We have also added the relevant literature that is in line with the subject of the manuscript. This can be found at pages 5-6 of the manuscript. We have also corrected the grammar mistakes throughout the paper.

Reviewer 4 Report

The article is written on a current topic. The authors made a good comparison of the material. Showed the perspective of the study. The authors give a sufficient description of the problem. The problem is not so global, but it exists and therefore the solution is interesting. There are few formulas and charts. There are no algorithms. Approaches to implementation are described quite well. Comparative indicators need to be improved. The bibliography is ok.

Author Response

Hi reviewer, thank you for comments and feedback, we have added tables to compare the related work and the results on pages 6, 12-14 of the manuscript. The algorithms/protocols were presented our other published work as it was not part of this article’s contribution, the references of these papers were cited in this manuscript. However, we added the parameters and metrics used in evaluating these underlying security protocols of the conceptual framework in this paper. We have also corrected the grammar mistakes throughout the paper.